# Exposure to (Poly)phenol Metabolites after a Fruit and Vegetable Supplement Intake: A Double-Blind, Cross-Over, Randomized Trial

**DOI:** 10.3390/nu14224913

**Published:** 2022-11-20

**Authors:** Cindy Romain, Letizia Bresciani, Jananee Muralidharan, Pedro Mena, Linda H. Chung, Pedro E. Alcaraz, Daniele Del Rio, Julien Cases

**Affiliations:** 1Innovation and Scientific Affairs, Fytexia, 34350 Vendres, France; 2Human Nutrition Unit, Department of Food & Drug, University of Parma, Via Volturno 39, 43125 Parma, Italy; 3Research Center for High Performance Sport, UCAM Universidad Católica de Murcia, 30107 Murcia, Spain

**Keywords:** (poly)phenol-based supplement, pharmacokinetics, urinary excretion, bioavailability, inter-individual variability

## Abstract

Dietary (poly)phenol intake derived from the daily consumption of five portions of fruits and vegetables could protect against the development of non-communicable diseases. However, the general population does not meet the recommended intake. Supplementation with (poly)phenol-rich ingredients, within a varied and balanced diet, could help in filling this nutritional gap. This study aimed to validate the proof-of-concept of a (poly)phenolic supplementation developed to enhance the daily consumption of potentially bioactive compounds. Oxxynea^®^ is a (poly)phenol-rich ingredient developed to provide the quantity and the variety corresponding to five-a-day fruit and vegetable consumption. In this double-blind, randomized cross-over study, 10 participants were supplemented with 450 mg of a (poly)phenol-based supplement or a placebo. Pharmacokinetics and urinary excretion profiles were measured for 24 and 48 h, respectively, using UPHLC-MS/MS analysis. The pharmacokinetic profile displayed a triphasic absorption, indicating peaks of circulating metabolites at 1.75 ± 0.25 h, 4.50 ± 0.34 h, 9.50 ± 0.33 h and an average T_max_ (time of maximal plasma concentration) of 6.90 ± 0.96 h. Similarly, the urinary profile showed maximum metabolite excretion at 3–6 h, 6–10 h and 14–24 h after supplement consumption. Compared to individual metabolites belonging to different (poly)phenolic subfamilies, the total circulating and excreted metabolites showed a reduced coefficient of variation (CV 38%). The overall bioavailability estimated was 27.4 ± 3.4%. Oxxynea^®^ supplementation may provide a sustained exposure to several (poly)phenolic metabolites and catabolites and reduces the inter-individual variation that could arise from supplementing only one class of (poly)phenol.

## 1. Introduction

The recognized protective effect of fruit and vegetable consumption on the incidence of non-communicable diseases (NCDs) such as cardiovascular diseases, type 2 diabetes mellitus and cancer has been confirmed in several recent meta-analyses [1,2,3]. According to the latest Global Burden of Disease study (GBD) [4], worldwide, 3.9 million deaths and more than 95 million disability-adjusted life-years (DALYs) were attributable in 2017 to a diet low in fruits and vegetables.

The health benefits of fruit and vegetable consumption have been ascribed to different bioactive nutrient molecules, such as dietary fibers, vitamins and minerals, and more recently, to non-nutritive phytochemicals, namely phenolic compounds, which are the most prominent and investigated substances [5]. Phenolics are of particular interest, as substantial epidemiological evidence shows inverse associations between their regular consumption and the incidence of certain NCDs [6,7,8].

Evidence generated through both animal and human intervention studies shows that the protective effects of phenolic compounds on NCDs may arise from several complementary mechanisms of the actions of (poly)phenols, including their capability to regulate gene expression, signaling pathways, metabolism and physiological responses [9,10].

However, it is noteworthy that despite the amount of evidence of the health benefits of phenolic compounds, which have even been discussed as “lifespan essentials” [11], and unlike vitamins and minerals, no dietary reference intake (DRI) has been set so far. The difficulty stems from the considerable diversity of phenolic compounds as, to date, more than 50,000 different phenolic structures have been identified [12]. Such structural variety is a determinant for their digestive metabolism and further absorption. Indeed, some compounds may be absorbed in the small intestine, where phase II enzymes metabolize them before entering systemic circulation [13], while the remaining unabsorbed compounds enter the large intestine, where they are degraded by the colonic microbiota, generating postbiotics of which a portion will, in turn, be absorbed and enter into the circulatory system [14]. Moreover, food matrix diversity and interindividual variabilities are also considered essential factors in pathways and the degrees of absorption of phenolics. Consequently, there is still a matter of intensive research to understand the correlation between the absorption of phenolic metabolites and their bioactivity. Thus, studying pharmacokinetics in systemic circulation and their pattern of urinary excretion is essential to have a comprehensive overview for the basis of their in vivo benefits.

Taking this complexity into account, to date, best (poly)phenol-related dietary guidelines have been established on the adequate consumption of fruits and vegetables. The World Health Organization (WHO) recommends a minimum daily intake of 400 g of fruits and vegetables [15]. Moreover, the WHO has stated that this intake must be varied at each meal to ensure an adequate intake of nutrients, including phenolic compounds [16]. In terms of (poly)phenol consumption, it is intriguing to hypothesize whether one of the rationales behind this recommendation is to allow sustained exposure to phenolic metabolites in circulation throughout the day. Based on these guidelines, the “5-a-day” fruit and vegetable campaign recommends consuming a varying number of 75–80 g servings of fruits and vegetables daily. Such guidelines are supported by several dose-response meta-analyses [1,17,18], among which the most recent included about 1.9 million participants worldwide [18]. The authors observed that the intake of five varied servings per day of fruits and vegetables was associated with lower mortality and that risk reduction plateaued at this daily amount.

It is noteworthy that 10 years after the start of the “5-a-day” promotion campaigns, the number of estimated deaths worldwide attributable to low fruit and vegetable consumption has doubled [19]. Recent reports aimed at examining the major fruit and vegetable campaigns and interventions conducted worldwide over the last 10 years concluded that the daily intake of fruits and vegetables remained well below the recommended WHO levels in both developed and developing countries [20,21].

Based on such observations, Oxxynea^®^ has been developed based on the “5-a-day” recommendations in terms of phenolic compound intake from a diversity of fruits and vegetables, as a possible strategy to increase the daily consumption of potentially bioactive compounds in the framework of a varied and balanced diet. In this regard, the botanical extracts have been selected for (1) their phenolic content, comparable to the phenolic quantity brought by the consumption of five fruits and vegetables a day; (2) their diversity in terms of phenolic structure varieties. A key criterion was the potential of enhancing the benefit derived from consuming a complex variety of fruits and vegetables at each meal that would allow a sustained release of a plethora of (poly)phenol metabolites in circulation throughout the day, having metabolites usually appearing at different times. This study was planned to overcome the consumption of a single variety of (poly)phenols and, accordingly, support a potential multi-phasic absorption.

Thus, this study aimed to validate the proof-of-concept of a (poly)phenolic supplementation developed to enhance the daily consumption of potentially bioactive compounds. The bioavailability of phenolic compounds following Oxxynea^®^ intake was evaluated through the absorption profiles and the urinary excretion of the supplemented (poly)phenols.

## 2. Materials and Methods

### 2.1. Development of a Nutritional Supplement Containing 5 Servings of Fruit and Vegetable (Poly)phenols

The ingredient, namely Oxxynea^®^, developed by FYTEXIA (France), is a food supplement that boasts a total (poly)phenol content comparable to the 5 most consumed fruits and vegetables [22], including apple, tomato, banana, orange and carrot.

The supplement was principally obtained by water/ethanol extraction of olive leaf (*Olea europea* L.), grape seed (*Vitis vinifera* L.), pomegranate (*Punica granatum* L.), grapefruit (*Citrus paradisi* Macfad.), orange (*Citrus sinensis* L. Osbeck) and by water extraction of green tea (*Camellia sinensis* L. Kuntze), bilberry (*Vaccinium myrtillus* L.) and white and red grape pomace (*Vitis vinifera* L.).

#### 2.1.1. Total Phenolic Content

The total phenolic content of the supplement was determined using the Folin–Ciocalteu method adapted to fruit and vegetable (poly)phenol quantitation according to Brat et al. [23], using gallic acid as the standard. The total phenolic content of the 5 selected benchmark servings of fruits and vegetables was calculated based on Brat et al. quantifications [23].

#### 2.1.2. Antioxidant Capacity

Both total antioxidant defences and total antioxidant reserves of the supplement were monitored in fresh heparinized blood using the KRL (Kit Radicaux Libres) and RESEDA (RESErves Défenses Antioxydantes) tests (Spiral Laboratories, Couternon, France), as previously described [24]. Briefly, the KRL test allows the evaluation of red blood cell resistance against 2,2′-azobis (AAPH)-induced free radicals and is measured as the time needed to haemolyze 50% of the red blood cells. The RESEDA test is based on the same principle, with a step of releasing biologically active and potentially antioxidant molecules in reserve. The KRL and RESEDA tests were also applied to the (poly)phenol fraction of a blend of 400 g of the 5 fruit and vegetable selected benchmark.

#### 2.1.3. HPLC Analysis of Phenolic Compounds in the Supplement

Phenolic compounds within the supplement were analyzed using two different analytical methods. The first approach corresponds to an internal HPLC (High-Performance Liquid Chromatography) analysis targeting a few key compounds to check for and ensure reproducibility between different production batches (quality control). Secondly, an exhaustive characterization and quantification of phenolic compounds from the supplement provided during the clinical trial was performed through ultra-High-Performance Liquid Chromatography (uHPLC) coupled with mass spectrometry (MS).

The HPLC internal method was carried out employing an Agilent HPLC 1260 apparatus (software Openlab CDS chemstation edition) coupled with a diode array detector. Separations were carried out by means of a Zorbax Stablevond SB-C18 column (4.6 × 1.5 mm; 5 µm particle size). In order to detect different phenolic classes, three different analytical methods were adopted: one for flavan-3-ols, flavanones and secoiridoids and phenylethanoids, one for anthocyanins and one for ellagitannins.

Regarding flavan-3-ols, flavanones and secoiridoids and phenylethanoids, mobile phase A consisted of 100% water, mobile phase B was 100% acetic acid and mobile phase C was 100% acetonitrile. The linear gradient program was used as follows: (a) 0 to 5 min 94% A and 6% B; (b) 5 to 10 min 82.4% A, 5.6% B and 12% C; (c) 10 to 15 min 76.6% A, 5.4% B and 18% C; (d) 15 to 25 min 67.9% A, 5.1% B and 27% C; (e) 25 to 30 min 65% A, 5% B and 30% C; (f) 30 to 40 min 100% C; (g) 40 to 45 min 64% A and 6% B. Monitoring was performed at 280 nm at a flow rate of 1 mL/min and injection volume of 25 µL. Flavan-3-ols were expressed as naringin or hesperidin, flavanones as catechin, epigallocatechin gallate epicatechin or epicatechin gallate equivalents and secoiridoids were expressed as oleuropein.

For anthocyanin analysis, mobile phase A was 100% water, mobile phase B consisted of 100% formic acid and mobile phase C was 100% acetonitrile. The linear gradient program was set as follows: (a) 0 to 5 min 84.2% A, 10% B and 5.8% C; (b) 5 to 20 min 77.6% A, 10% B and 12.4% C; (c) 20 to 35 min 68.2% A, 10% B and 21.8% C; (d) 35 to 40 min 58.8% A, 10% B and 31.2% C; (e) 40 to 45 min 44.7% A, 10% B and 45.3% C; (f) 45 to 50 min 44.7% A, 10% B and 45.3% C; (g) 50 to 60 min 40% A, 10% B and 50%C; (h) 60 to 65 min 84.2% A, 10% B and 5.8% C. Monitoring was performed at 520 nm at a flow rate of 0.8 mL/min and injection volume of 10 µL. Anthocyanins were expressed as kuromanin equivalent.

Concerning ellagitannins, mobile phase A was 100% water, mobile phase B was 100% phosphoric acid and mobile phase C was 100% methanol. The linear gradient program was: (a) 0 to 5 min 94.9% A, 0.1% B and 5% C; (b) 5 to 15 min 89.9% A, 0.1% B and 10% C; (c) 15 to 30 min 69.9% A, 0.1% B and 30% C. Monitoring was performed at 260 nm at a flow rate of 0.8 mL/min and injection volume of 5 µL. Ellagitannins were expressed as punicalagin equivalent.

Regarding exhaustive characterization of phenolic compounds from the supplement, the dried powder was firstly extracted in triplicate, as reported by Mena et al. [25]. Briefly, 200 mg of powder was extracted with 1 mL of 80% aqueous methanol acidified with formic acid (1%) and sonicated for 25 min. The mixture was centrifugated at 12,000 rpm for 10 min at room temperature, and the supernatant was collected. Two additional extractions were performed for each sample with an extra 0.5 mL of the same solvent, as described above, after which they were centrifugated. The three supernatants were pooled before uHPLC-MSn analysis.

The methanolic extract was analyzed using an Accela uHPLC 1250 apparatus equipped with a linear ion trap (LIT) MS (LTQ XL, Thermo Fisher Scientific Inc., San Jose, CA, USA) fitted with a heated ESI (H-ESI-II) probe (Thermo Fisher Scientific Inc.). Separation was carried out by means of a Restek C18 (100 × 2.1 mm) column, 3 μm particle size (Restek Corporation, Bellefonte, PA, USA).

Based on previous works [26,27], anthocyanins were detected in positive ionization mode, with mobile phase, pumped at a flow rate of 0.3 mL/min, consisting of a mixture of acidified acetonitrile (0.1% formic acid) (solvent A) and 0.1% aqueous formic acid (solvent B). Following 0.5 min of 5% solvent A in B, the proportion of A was increased linearly to 51% over a period of 8.5 min. Solvent A was increased to 80% in 0.5 min, maintained for 2 min, then the start condition was re-established in 0.5 min and maintained for 5 min to re-equilibrate the column (total run time: 17 min). The H-ESI-II interface was set to a capillary temperature of 275 °C, and the source heater temperature was 300 °C. The sheath gas (N2) flow rate was set at 40 (arbitrary units), and the auxiliary gas (N2) flow rate was at 5. During anthocyanin analysis, the source voltage was 4.5 kV, and the capillary and tube lens voltage were 20 and 95 V, respectively.

For all other phenolic classes, spectrometric analyses were performed in negative ionization mode. Maintaining the same chromatographic gradient, the H-ESI-II interface was set to a capillary temperature of 275 °C, and the source heater temperature was 250 °C. The sheath gas (N2) flow rate was set at 60 (arbitrary units), and the auxiliary gas (N2) flow rate at 15. The source voltage was 4 kV, the capillary voltage was −49 V and the tube lens voltage was −153 V. A preliminary analysis of 5 μL of samples was carried out using full-scan, data-dependent MS3 analysis, to unequivocally identify the aglycones, scanning from a mass to charge (*m*/*z*) range of 100–900 for anthocyanins and from *m*/*z* range 100–2000 for the negative analysis, using a collision-induced dissociation (CID) equal to 35 (arbitrary units) to obtain fragmentation.

Data processing was performed using Xcalibur software from Thermo Scientific (Thermo Fisher Scientific Inc.). All compounds were quantified through external calibration with commercial standards, when available, or with the most structurally similar reference compound (Appendix A).

Chemicals and reagents used for characterization are available in Appendix A.

### 2.2. Pharmacokinetics and Urinary Excretion

#### 2.2.1. Subjects

Ten healthy participants were recruited from the region of Murcia in southern Spain to participate in the study, in February 2018. Both men and women, 18 years old or older, having a normal body mass index (BMI) range (18.50–24.99 kg/m^2^), were included in the study. Subjects were excluded if they had a metabolic disorder or any kind of disease, were using medication or food supplements, were pregnant or breastfeeding, were smokers or had an allergy to any component of the supplement.

The study was approved by the UCAM (Universidad Católica San Antonio de Murcia, Spain) Ethics Committee (EC-number: CE111701) and conducted per guidelines laid out in the Declaration of Helsinki [28] and in compliance with Good Clinical Practices defined in the ICH Harmonized Tripartite Guideline [29]. All participants were informed about the study procedures, agreed to adhere to lifestyle modifications throughout the course of the study and signed written informed consent before entering the study. This trial was registered at clinicaltrials.gov as NCT03432104.

#### 2.2.2. Study Design

This study was designed as a 3-week randomized, double-blind, cross-over clinical trial. Eligible participants were randomized using simple block randomization of 1:1. Once enrolled, subjects received either one 450 mg capsule of the test supplement or one 450 mg capsule of placebo on the first day of the study (Phase I: D1). After three days of sampling, volunteers started a wash-out period of 1 week and were then enrolled for the second test Phase with ingestion of the opposite supplement (Phase II: D1) (Figure 1). The placebo product was 100% maltodextrin, which was (poly)phenol-free.

Both the supplement and the placebo were supplied in a 450 mg capsule of identical opaque appearance and flavor.

Two days before the supplementation and during both Phase I and Phase II, volunteers were asked to consume a (poly)phenol-free diet. In order to facilitate participant adherence to this dietary restriction, the Research Center provided meals and snacks for the whole period, as well as a list of permitted and forbidden foods in order to avoid any deviations. Additionally, volunteers were asked to abstain from physical exercise for four days before the beginning of the study and throughout the study. Physical activity abstinence was monitored by an accelerometry device, the Fitbit^®^ activity tracker (Fitbit Inc., San Francisco, CA, USA).

#### 2.2.3. Plasma Pharmacokinetics

On the day of the test (D1), volunteers visited the Research Center after an overnight fast. Blood was collected at T0 to determine baseline levels of plasma phenolic metabolites. Then, volunteers were invited to take the 450 mg capsule of the supplement or the placebo with 250 mL of still water (Figure 2). Additional blood samples were then collected at 1 h (D1, T1), 2 h (D1, T2), 3 h (D1, T3), 4 h (D1, T4), 5 h (D1, T5), 6 h (D1, T6), 8 h (D1, T8), 10 h (D1, T10) and 24 h (D2, T24) after capsule intake. Blood was taken from the basilica vein and centrifuged at 3000 rpm for 10 min at 4 °C. Plasma was immediately extracted and frozen at −80 °C for further analysis.

Plasma samples were then extracted using a solid phase extraction (SPE) method, as previously reported by Castello and colleagues [30]. Briefly, 350 µL of plasma samples was diluted (1:1) with phosphoric acid 4% (*v*/*v*). After plate activation, 600 µL of the diluted plasma samples was loaded on a 96 well µ-SPE HLB (Oasis^®^ HLB µElution Plate 30 µm, Waters, Milford, Massachusetts, MA, USA). Samples were washed with 200 µL of water and 200 µL of 0.2% (*v*/*v*) acetic acid. Finally, samples were eluted with 60 µL of methanol for UHPLC-ESI-MS/MS analysis.

#### 2.2.4. Urinary Excretion

Urine samples were collected the day before (baseline) and during the 48 h period following the supplementation (T0-T3, T3-T6, T6-T10, T10-T14, T14-T24, T24-T32 and T32-T48) (Figure 2). Urine volumes at each collection period were registered, and samples were aliquoted and stored at −80 °C until further processing.

Then, urine samples were prepared as previously reported by Brindani and colleagues [31]. Briefly, urine samples were defrosted, vortexed, diluted in 0.1% formic acid in water (1:4, *v*/*v*) and centrifuged at 12,000 rpm for 5 min. Finally, urine samples were filtered (0.45 µm nylon filter) prior to UHPLC-ESI-MS/MS analysis.

#### 2.2.5. Plasma and Urine UHPLC-ESI-MS/MS Analysis

Plasma and urine samples were analyzed by a UHPLC DIONEX Ultimate 3000 equipped with a triple quadrupole TSQ Vantage (Thermo Fisher Scientific Inc) fitted with a heated ESI (H-ESI) (Thermo Fisher Scientific Inc) probe, as previously reported [31]. Separations were carried out by means of a Kinetex EVO C18 (100 × 2.1 mm) column, 2.6 µm particle size (Phenomenex, Torrance, CA, USA), installed with a pre-column (Phenomenex). Mobile phase A was 0.2% formic acid in water, and mobile phase B was acetonitrile containing 0.2% formic acid. The gradient started with 5% B, and isocratic conditions were maintained for 0.5 min, reaching 95% B after 6.5 min, followed by 1 min at 95%. The starting gradient was then immediately re-established and maintained for 5 min to re-equilibrate the column. The flow rate was 0.4 mL/min, the injection volume was 5 µL and the column temperature was set at 40 °C.

The applied MS method consisted of the selective determination of each target precursor ion by the acquisition of characteristic product ion in selective reaction monitoring (SRM) mode (Appendix A). For all the analyses, the spray voltage was set at 3 kV, the vaporizer temperature at 300 °C and the capillary temperature operated at 270 °C. The sheath gas flow was 50 units, and auxiliary gas pressure was set to 10 units. Ultra-high purity argon gas was used for collision-induced dissociation (CID). The S-lens values were defined for each compound based on infusion parameter optimization. Conversely, for those compounds not available for infusion, the S-lens values were set using the values obtained for the chemically closest available standards. Quantification was performed with calibration curves of standards when available or using the most structurally similar compound. Data processing was performed using Xcalibur software (Thermo Scientific Inc., San Jose, CA, USA).

Chemicals and reagents used for metabolite and catabolite quantification are available in Appendix A.

#### 2.2.6. Data Analysis

All data were expressed as mean values ± standard deviation (SD) and mean values ± standard error to the mean (SEM) for metabolites in blood and urine samples. PKSolver add-on program [32] in Microsoft Excel was used to determine pharmacokinetic parameters of blood phenolic metabolites, including time to reach the maximum plasma concentration (Tmax, h), maximum plasma concentration (Cmax, nmol/L), area under the curve (AUC0-24, nmol/L·h), and time of presence of metabolites in circulation (Tpres, h). Tpres was calculated by taking the time interval between the first appearance and the last appearance of the considered metabolite. Coefficient of variation (CV) was evaluated for each metabolite and multivariate principal component analysis (PCA) was applied to explore the inter-individual variation in the urinary metabolites among the participants. Principal component analysis was performed in R software (v 4.1) using the package mixomics (v 6.16.3) [33].

## 3. Results

### 3.1. Total Phenolic Content and Antioxidant Capacity

In the selected 400 g of the five most consumed fruits and vegetables in France, the total phenolic content was 228 mg gallic acid equivalent (GAE). Therefore, the supplement was developed to provide the same quantity of total (poly)phenols, which was confirmed by the Folin–Ciocalteu analysis. One daily dose of 450 mg of the test supplement provided 230 mg of GAE total phenolic compounds (Table 1). Following the same approach, the antioxidant capacity of the supplement was similar to the value calculated for 400 g of the five fruit and vegetable references, corresponding to 383 mg GAE and 396 mg GAE, respectively (Table 1).

### 3.2. HPLC Analysis If Phenolic Compounds in the Supplement

The HPLC internal analysis of the supplement revealed five main families of phenolic compounds found in the supplement, including, in decreasing order, flavan-3-ols > flavanones > ellagitannins > secoiridoids and phenylethanoids > anthocyanins. The total phenolic content of the supplement was established at 144.8 ± 9.9 mg (Table 2).

The more comprehensive analysis, performed through uHPLC-ESI-MSn, allowed us to identify or tentatively identify 136 phenolic compounds, among which 124 were quantified (Table 3). Phenolic compounds belonged to different phenolic families, i.e., in a decreasing order, flavan-3-ols (35.2%) > ellagitannins (19.2%) > phenylethanoids (14.8%) > flavanones (12.6%) > secoiridoids (9.0%) > flavones (3.4%) > flavonols (2.6%) > hydroxybenzoic acids (1.7%) > anthocyanins (0.8%) > hydroxycinnamic acids (0.3%) > stilbenoids (0.2%) > gallotannins (0.1%) > dihydrochalcones (<0.1%). The total phenolic content of the supplement was established at 221.1 ± 9.5 mg, corresponding to 594.9 ± 14.5 µmol (Table 3).

### 3.3. Characteristics of the Subjects

The selected participants were six males and four females with the following characteristics: age 27.8 ± 5.5 years, height 173.1 ± 9.0 cm, body weight 69.0 ± 11.0 kg and BMI 22.9 ± 1.8 kg/m^2^. All participants followed the recommendations in terms of abstinence from physical activity as assessed with the Fitbit^®^ activity tracker (data not shown).

### 3.4. Plasma Pharmacokinetics

The absorption pattern of total plasma metabolites in the supplemented group revealed a 24 h higher presence of circulating phenolic metabolites and clearly displayed three distinct absorption phases throughout the gastrointestinal tract, showing a first absorption peak occurring 2 h after taking the supplement, a second peak arising 5 h post-consumption and a last peak observed 10 h after supplement consumption. Out of the 47 monitored phenolic metabolites, 28 different plasma phenolic metabolites were identified and quantified (Table 4 and Appendix A), belonging to 7 different classes of phenolic metabolites, according to the (poly)phenol classes identified in the food supplement. All individual metabolites were mainly found as conjugated forms with sulfate, glucuronide, methyl or glycine moieties.

The relative contribution of the total AUC0-24 of each class of phenolic derivatives is presented in Figure 3. 4′-Hydroxyhippuric acid was not included in the analysis as this compound may originate from both phenolic metabolism and endogenous precursors [34].

Two main subclasses of flavan-3-ol derivatives were identified: (epi)catechin phase-II conjugates and phenyl-y-valerolactone and phenylvaleric acid derivatives. Among them, the second subclass was predominant in blood with a relative AUC0-24 eight times more abundant than the (epi)catechin derivatives.

The three identified and quantified (epi)catechin conjugates (Id. 3–5) started appearing in the blood in the first two hours post-consumption (average Tmax = 3.1 ± 0.37 h), indicating an absorption starting in the upper part of the gastrointestinal tract. We observed inter-individual variability in both the absorption time and plasma concentration (AUC0-24) of these compounds. The variation of AUC0-24, expressed as CV, ranged from 48% for (epi)catechin-glucuronide (isomer 2) to 89% for methoxy-(epi)gallocatechin-glucuronide. Considering the total AUC0-24 of (epi)catechin derivatives, the variation in AUC dropped to 40%. The 12 detected phenyl-y-valerolactone and phenylvaleric acid derivatives (Id. 9, 10, 13–18, 20–23) appeared later in the circulatory system, peaking between 6 and 24 h for most volunteers. The average Tmax was 8.50 ± 2.30 h post-intake, confirming the predominant colonic origin of these compounds. Consequently, the presence of these compounds was extended to 21.6 ± 0.72 h. It is noteworthy that the inter-individual variability was much higher for these compounds, as AUC0-24 varied from 84% to 144%; however, considering the sum of phenyl-y-valerolactone and phenylvaleric acids, the CV of total AUC0-24 was reduced to 90%.

Flavanone derivatives accounted for more than 10% of the total AUC0-24 and displayed a biphasic absorption pattern. Indeed, naringenin and hesperetin derivatives (Id. 26–29) started appearing in the bloodstream in the first two hours post-ingestion of the supplement, indicating an absorption in the upper gastrointestinal tract. Nevertheless, the absorption pattern also demonstrated more important secondary peaks between 3 and 10 h post-consumption, involving a significant implication of colonic microbiota in the absorption and metabolism of the flavanone derivatives. Such a biphasic absorption promoted an average presence time in the bloodstream of 21.65 ± 0.42 h.

Only one phenylethanoid derivative, namely 2-(phenyl)ethanol-3′-glucuronide (Id. 33), was quantified in the plasma with the analytical method employed. This compound was rapidly absorbed (Tmax 1.45 ± 0.32 h) with the highest AUC in the first two hours post-intake. The inter-individual variability in AUC0-24 of this metabolite was 36%.

Hydroxybenzoic acids and simple benzenes (Id. 36, 37, 45, 47) were the most abundant class of phenolic metabolites, accounting for nearly 45% of total AUC0-24. Excluding 4-hydroxyhippuric acid, dihydroxybenzene-sulfate (Id. 36) presented the highest Cmax, AUC0-24 and persistence in the blood for this class of catabolite. This compound derives from gut microbiota metabolism, as confirmed by Tmax 7.10 ± 0.86 h. 4-Hydroxybenzoic acid-3-sulfate (Id. 45) also presented a biphasic absorption, with a first major peak occurring at 1.38 ± 0.18 h post-absorption, and a second minor peak between 3 and 10 h post-intake. The inter-individual variability of total AUC0-24 for this family of compounds was 54%.

Finally, hydroxyphenylpropanoic acid derivatives (Id. 40, 41, 46) represented 34.3% of the total AUC0-24. Three different derivatives of this class were identified and quantified, with 3-(3′-hydroxyphenyl)propanoic acid being the most abundant. These derivatives were microbial-derived metabolites with a Tmax arising between 4 and 7 h post-intake of the supplement. Also, AUC0-24 CV ranged from 129 to 301%, indicating high inter-individual variability in the production of these compounds.

Finally, the total metabolite concentration reached within the 24 h blood-sampling period is presented in Figure 4. The placebo group, as compared to the supplemented group, displayed a minimal absorption rate, thus confirming a good adherence to the (poly)phenol-free diet restriction. Compliance with this diet is also validated through the baseline quantification of metabolites (T0 h) that is <200 nmol/L within the two groups. It must be highlighted that mammalian metabolic pathways, including the hepatic metabolism of the surplus aromatic amino acids, exist and lead to low molecular weight catabolites, interfering with the quantified metabolites [35]. Considering the whole (poly)phenol plasma concentration quantified after supplement consumption, a triphasic absorption pattern can be outlined, including a first plasma concentration peak at 1.75 ± 0.25 h, accounting for phase II human metabolites, a second peak at 4.50 ± 0.34 h, accounting for the earlier microbial catabolites, and a last plasma concentration peak at 9.50 ± 0.33 h, described by those catabolites derived from a later microbiota activity (Figure 4). In general, the presence time (Tpres) of metabolites in the circulation was equal to 23.80 ± 0.24 h, the mean Tmax was 6.90 ± 0.96 h, and the total AUC0-24 was more than 10 µmol/L·h, with a CV of 58%.

### 3.5. Urinary Excretion

In urine, eight different classes of phenolic compounds were represented by forty-four different conjugated metabolites that were identified and quantified (Table 5 and Appendix A). The most abundant class identified was phenyl-y-valerolactone and phenylvaleric acid derivatives, accounting for more than 66% of total excretion, followed by hydroxybenzoic acids and simple benzenes (23.2%), hydroxyphenylpropanoic acids (6.1%), flavanones (1.9%), ellagitannins (1.5%), (epi)catechin derivatives (0.8%), phenylethanoid derivatives (0.4%) and other minor flavonoids (0.1%) (Figure 5).

Flavan-3-ol derivatives were the most abundant compounds quantified, with the class of phenyl-y-valerolactone and phenylvaleric acid derivatives being predominant. Three (epi)catechin derivatives identified in plasma were also recovered in urine with an excretion peak of 3 and 6 h post-consumption, confirming a predominant absorption in the upper gastrointestinal tract (Figure 6A). Inter-individual variability in the excretion of these derivatives was high, as CV ranged from 71% to 164%. A total of 17 different phenyl-y-valerolactone and phenylvaleric acid derivatives were identified in urine, and among them 12 were also detected in plasma (Id. 9, 10, 13–18, 20–23). Many of these compounds presented a double excretion peak, the first one between 6 and 10 h and the second one appearing at 14–24 h, indicating a preferential microbiota-derived metabolism along the large intestine (Figure 6B). Inter-individual variability, reported as CV, was, averagely, 42%.

Regarding flavanones, up to five different compounds were found in urine; naringin-diglucuronide (Id. 25) was the only flavanone derivative not detected in plasma. The excretion pattern of these derivatives confirms the biphasic absorption, showing, for most compounds, two distinct excretion peaks corresponding to 6–10 h and 14–24 h post-supplement consumption (Figure 6C). Also, for flavanone metabolites, a high inter-individual variability can be suggested (61%).

In contrast to plasma, where only one phenylethanoid derivative was identified, in urine both 2-(phenyl)ethanol-3′-glucuronide (Id. 33) and oleuropein-sulfate (Id. 34) were quantified. However, the excretion rate of the last one did not exceed 0.10 µg in 48 h. In general, both phenylethanoid derivatives were rapidly excreted with a peak occurring 3–6 h post-absorption, and their excretion was completed in 32 h (Figure 6D). In contrast to other metabolites, the inter-individual variability of these compounds was comparatively low (35%).

Hydroxyphenylpropanoic acids were the third most important class of compounds excreted. Two derivatives were quantified (Id. 40, 41) in urine samples, displaying a biphasic excretion pattern (Figure 6E) and a high inter-individual variability with a CV of 75%.

Hydroxybenzoic acids and simple benzenes were the second most abundant classes of urinary metabolites, mainly represented by trihydroxybenzene derivatives (Id. 36, 37), for which excreted concentration reached more than 4 mg after 48 h. These catabolites reached their maximum excretion rate between 14 and 24 h post-consumption, confirming their microbial origin (Figure 6F). The coefficient of variation for the excretion of this class of compounds was 39%.

Finally, three ellagitannin catabolites were quantified in urine, including 8-hydroxy-urolithin-3-glucuronide (Id. 42), 9-hydroxy-urolithin-3-glucuronide (Id. 43) and urolithin-3-glucuronide (Id. 44). It is noteworthy that six volunteers of nine excreted 8-hydroxy-urolithin-3-glucuronide, and, among them, two excreted urolithin-3-glucuronide as well (data not shown). The peak excretion of these ellagitannin catabolites resulted between 14 and 24 h, confirming their microbial origin (Figure 6G).

The total amount of urinary excreted metabolites over a 48 h period after supplement consumption is presented in Figure 6H. The placebo group, as compared to the supplemented group, displayed a minimal excretion rate. On the contrary, after supplement consumption, the urinary excretion pattern of (poly)phenol metabolites displayed two main peaks, confirming the different absorption steps within the gastrointestinal tract. After 48 h of supplement ingestion, the metabolite excretion result was four times higher than the curve recorded in the placebo group, indicating a sustained and more protracted release of various phenolic catabolites. The total urinary excretion of metabolites derived from supplement consumption was 43.0 mg, corresponding to 163.1 ± 20.1 µmol. Thus, based on the excreted µmols of quantified metabolites, the total bioavailability of phenolic compounds was evaluated to be 27.4 ± 3.4%.

### 3.6. Inter-Individual Variability

A high inter-individual variability was highlighted both in circulating plasma phenolic metabolites and in their urinary excretion (Table 4 and Table 5). The coefficient of variation of excreted (poly)phenol subfamilies ranged from 34.5% for phenylethanoid derivatives to 126% for ellagitannin derivatives. Considering the total excreted metabolites, the mean CV was 38%.

Unsupervised multivariate analysis (PCA) was applied to urinary metabolites measured at different collection points (from 0 to 48 h). Three principal components (PCs) explained up to 52% of the total variance. PC1 explained 29% of total variability and was mainly loaded by dihydroxyphenyl-y-valerolactone and dihydroxyphenylvaleric acid derivatives (Figure 7D). PC2 and PC3 explained 14% and 9%, respectively, of the total variability. PC2 presented positive loads for (epi)catechin derivatives and negative ones for urolithin derivatives. Contrary to PC1, PC3 was positively loaded mainly by monohydroxypehnyl-valerolactone, monohydroxyphenyl valeric acid derivatives, hydroxyphenylpropanoic acid derivatives and hydroxybenzoic acid derivatives (Figure 7D). Urine samples collected 10 h or more post-intake presented positive PC1 score values (Figure 7A) (Id. 21, 15, 13, 9). Samples collected during the 3 to 10 h urinary collection period had the highest positive score values for both PC2 and PC3 (Figure 7B). In general, the large inter-subject variability observed for metabolite production is well represented by PCA plots (Figure 7), the area covered by the surface delimited by time points being quite different among subjects.

## 4. Discussion

Strong evidence recommends increasing fruit and vegetable intake for health benefits. However, their consumption has not risen in recent years. Unarguably, nutritional supplements cannot replace the consumption of fruits and vegetables, which must remain the first option because of their higher nutritional complexity, which includes fiber, vitamins and minerals, besides bioactive compounds. However, a formulated food supplement rich in (poly)phenol compounds could aid in filling this nutritional gap. In this context, supplementation, inserted in a varied and balanced diet, represents a possible solution to improve the daily consumption of dietary (poly)phenols to respond to consumer demand, enhancing the well-known benefits of regular and varied fruit and vegetable consumption.

Based on the modified Folin–Ciocalteu method [23], the supplement provided the same amount of phenolic compounds as the selected references, i.e., the five most consumed fruits and vegetables in France. Our exhaustive analysis confirmed that the main (poly)phenolic families were represented, covering up to 124 different phenolic compounds. The most abundant subfamily in the supplement was flavan-3-ols, being that these phenolics are the most abundant both in grape and green tea extracts [36,37]. Ellagitannins, whose main compounds were ellagic acid and punicalagin derivatives, are characteristic of pomegranate extracts [38], whereas citrus extracts provided mainly flavanones comprising naringenin and hesperetin derivatives [39]. Secoiridoids and phenylethanoids are common phenolics of olive leaf extract [40], and anthocyanins mainly result from bilberry, grape and pomegranate extracts [38,41,42].

From the pharmacokinetic study, a triphasic absorption profile was obtained for circulating plasma metabolites, indicating that some compounds are rapidly absorbed in the upper part of the gastrointestinal tract, although the most extensive part necessitates being firstly metabolized by the colonic microbiota before absorption. Such a pattern validates that the intake of the supplement helps provide a continuous supply of various metabolites over a time period of 24 h that corresponds to a continuous absorption. The global pattern of the urinary excretion profile showed that excreted phenolic metabolites had not yet reached back baseline level 48 h post-consumption, resulting in a long-lasting phenolic presence in the bloodstream. This supports the hypothesis that mimicking the main meal intake of five fruits and vegetables would deliver a sustained and continuous release of metabolites. However, there are limited studies that have evaluated the circulating metabolites after the recommended intake of fruits and vegetables over a 24 h period [43]. This gap in the literature has been noted in a few reviews, which have indicated the need for such studies in the future [44,45], especially focusing on understanding the time-dependent distribution of metabolites in the systemic circulation, as has been recently reported for coffee consumption [46], for instance.

A total of 28 different circulating metabolites and up to 44 urine-excreted compounds were identified and quantified, among which flavan-3-ol-derived metabolites covered more than half. Among the last, 3 phase II (epi)catechin derivatives were identified early in plasma, indicating an absorption starting in the first part of the gastrointestinal tract, in accordance with previous reports [47]. These metabolites arise from the conjugation of parent compounds at both the enterocyte and hepatic levels, generating methylated, glucuronidated and sulfated (epi)catechin derivatives in the bloodstream [48]. However, the main flavan-3-ol parent compounds are metabolized by the gut microbiota, generating characteristic flavan-3-ols colonic catabolites, phenyl-y-valerolactones and, by breaking the valerolactone ring, phenylvaleric acids [49,50]. Here, 12 different phenyl-y-valerolactones and phenylvaleric acids were quantified in plasma with an average Tmax of 8.50 ± 2.30 h, confirming the colonic origin of these compounds. It is noteworthy that further metabolizing these compounds through β-oxidation of the side chain of the phenylvaleric acids also gives rise to lower molecular weight phenolic acids such as phenylpropanoic and hydroxybenzoic acid catabolites [50], which were both identified with relatively high AUC in the present pharmacokinetic study. However, these phenolic acids are also gut microbial metabolites of other flavonoids [49,51], as well as from the hepatic metabolism of the surplus aromatic amino acids [35], thus explaining their high concentration in the blood. The excretion pattern of flavan-3-ols metabolites confirmed the extensive bioconversion of the parent compounds at both intestinal and colonic levels.

The metabolism of flavanones demonstrated a typical pattern of absorption and excretion depicted by a biphasic curve, as previously shown [34,51,52]. It is noteworthy that colonic metabolism is, however, predominant in the formation of flavanone metabolites, as demonstrated by an average Tmax of 5.30 ± 1.17 h and confirmed by a delayed urinary excretion time. On the contrary, olive-derived phenolic compounds were rapidly absorbed and excreted, indicating a predominant role of the small intestine in their metabolism, and confirming that oleuropein is an effective source of hydroxytyrosol [53]. Finally, three ellagitannin derivatives were quantified in urine, namely 8-hydroxy-urolithin-3-glucuronide, 9-hydroxy-urolithin-3-glucuronide and urolithin-3-glucuronide, typical metabolites derived from ellagic acid lactone ring cleavage, decarboxylation and dehydroxylation reactions [54].

A key point to be considered in this study is the high inter-individual variability in both the absorption and excretion of phenolic compounds. Such variability may be driven by several factors, including genetic makeup, gut microbiota composition and functionality, age, gender and physiological status [55]. As an example, genetic variants in the catechol methyl transferase (COMT) [56] and the sulfotransferase (SULT) [57] enzymes may explain, at least in part, the variability illustrated in the excretion of native forms of metabolites, such as (epi)catechin derivatives (Id. 5, 6, 7). It is noteworthy that colonic metabolites displayed an even higher inter-individual variability, emphasizing the key role of the gut microbiota in (poly)phenol catabolism and bioavailability. Emerging research in this field describes the existence of different (poly)phenol-metabolizing phenotypes, namely metabotypes that have been so far clearly described for soya isoflavones and ellagitannins [58,59,60]. The latter is a good illustration of the impact of the microbiota; indeed, individual data (not shown) demonstrated that six volunteers (amongst nine) in our study were able to produce 8-hydroxy-urolithin-3-glucuronide, and that, among them, only two were able to additionally produce 9-hydroxy-urolithin-3-glucuronide and urolithin3-glucuronide. This is in agreement with the distribution of the three urolithin metabotypes previously defined [61]. Moreover, the inter-individual variability in flavan-3-ol production based on quali-quantitative differences, instead of the production or non-production of specific catabolites, which emerged in the present work, is in accordance with recent evidence on the existence of metabotypes of flavan-3-ol colonic metabolites [50,62].

Obviously, depending on the metabotype, the inherent nutritional benefits attributable to the resulting bioactive phenolic metabolites or catabolites may greatly vary between subjects [63]. Considering this main observation, this study aimed at counterbalancing the inter-individual variability by providing a sufficiently varied source of phenolic compounds from several plant origins to compensate for and balance individual gaps. Indeed, as hypothesized earlier, a low CV for total metabolites was observed after supplement consumption, indicating the ability of various participants to utilize distinct substrates differently. Potential individual benefits may change among individuals, of course, but considering the pleiotropic effects of (poly)phenols in disease prevention this should not be a major issue. The observation of low inter-individual variability demonstrates that eating a diversity of (poly)phenols from several fruit and vegetable sources, as should be the case if the five-a-day recommendation is followed, must help to improve the occurrence and absorption of diverse bioactive metabolites providing nutritional benefits. This is in contrast to the consumption of individual (poly)phenols or not-well-characterized botanical sources, for which some individuals may be unable or limited in metabolizing parent compounds, resulting in “non-responder” putative nutritional benefits. However, a limit of the study is the restricted controlled diet followed by the volunteers. Although a free-(poly)phenol diet is needed to establish the metabolism, absorption and bioavailability of the (poly)phenol supplement, it would be reasonable to evaluate the actual influence of supplement consumption in daily (poly)phenol intake and in circulating bioactive metabolites in the context of a varied and balanced diet.

Finally, the bi-directional relationship existing between (poly)phenols and the gut microbiota must also be taken into consideration [57]. Indeed, chronic supplementation with phenolic compounds can modulate the microbiota composition of individuals, impacting the inter-individual variability in their absorption, metabolism and bioavailability, resulting in modified and hopefully enhanced nutritional benefits.

The question of the diversification of the source of fruits and vegetables has been set as a better predictive factor for nutritional benefits. Accordingly, future investigations should focus on nutritional endpoints linked to the well-known benefits associated with a regular consumption of varied (poly)phenolic compounds from fruits and vegetables. This would include an investigation of the possible links related to the inter-individual variation in biomarkers of the physiological or biological parameters of interest.

## 5. Conclusions

In conclusion, this randomized, double-blind, cross-over clinical trial study demonstrated that the supplementation of a complex (poly)phenolic ingredient could overcome the issue of inter-individual variation in bioavailability, linked to the ability of subjects to metabolize substrates differently. Contemporary consumption of a (poly)phenol-rich supplement, in the context of a varied and balanced diet, may benefit individuals in improving the daily consumption of bioactive compounds and, consequently, the occurrence and absorption of diverse bioactive metabolites, providing long exposure to bioactive circulating molecules. Future studies are warranted to understand the chronic effects of complex (poly)phenolic ingredient supplementation on biological biomarkers and its bioactivity and bioefficacy.

## Figures and Tables

**Figure 1 nutrients-14-04913-f001:**
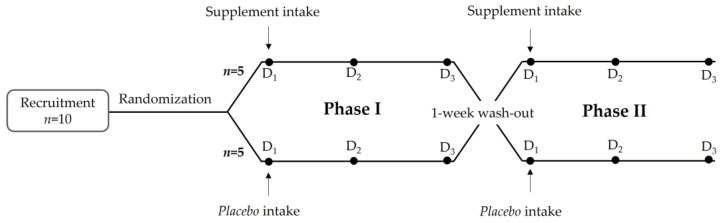
General flow diagram of the study.

**Figure 2 nutrients-14-04913-f002:**
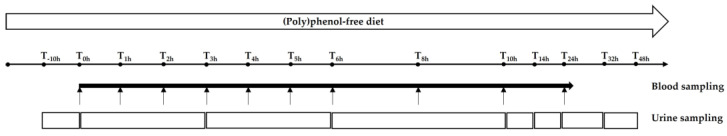
Blood and urine collection time during the 48 h period of both phase I and phase II.

**Figure 3 nutrients-14-04913-f003:**
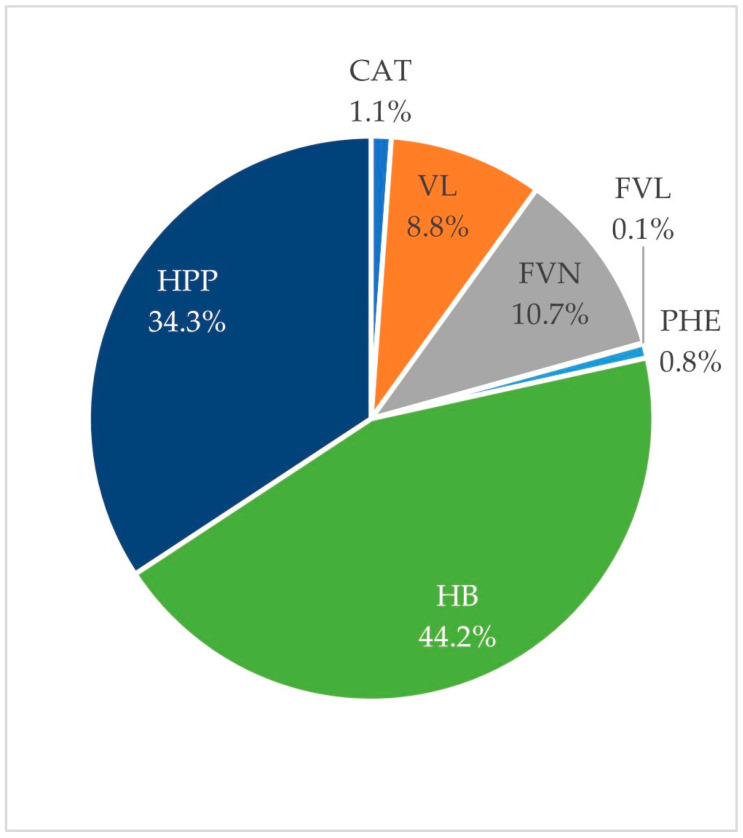
Relative plasma AUC0-24 for each class of phenolic metabolite. HB: Hydroxybenzoic acids and simple benzenes; HPP: Hydroxyphenylpropanoic acids; FVN: flavanone derivatives; VL: Phenyl y-valerolactones and phenyl valeric acids; CAT: (Epi)catechin derivatives; PHE: Phenylethanoid derivatives; FVL: Flavonol derivatives. 4′-hydroxyhippuric acid was not included.

**Figure 4 nutrients-14-04913-f004:**
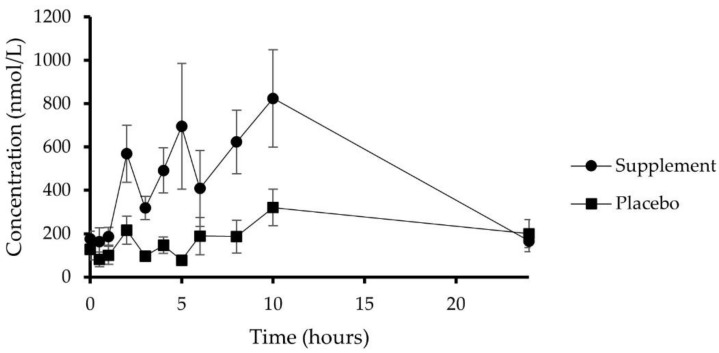
Sum of all (poly)phenol metabolites detected in plasma in placebo (*n* = 10) and supplemented subjects (*n* = 10) over a 24 h period. Data are expressed as means ± SEM.

**Figure 5 nutrients-14-04913-f005:**
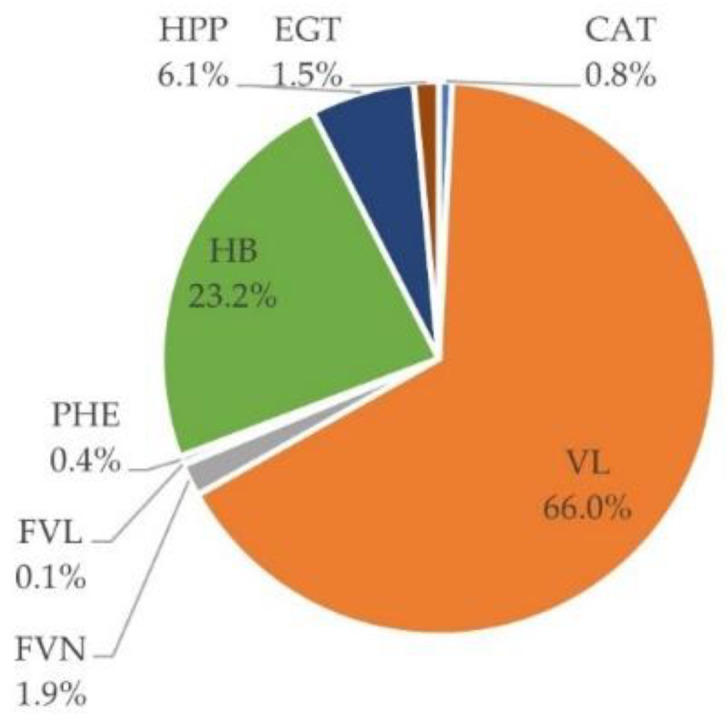
48 h urinary excretion of phenolic metabolites. HB: Hydroxybenzoic acids and simple phenols; HPP: Hydroxyphenylpropanoic acids; FVN: Flavanone derivatives; VL: Phenyl- y-valerolactones and phenyl valeric acids; CAT: (Epi)catechin derivatives; PHE: Phenylethanoid derivatives; FVL: Flavonol derivatives. 4′-hydroxyhippuric acid was not included.

**Figure 6 nutrients-14-04913-f006:**
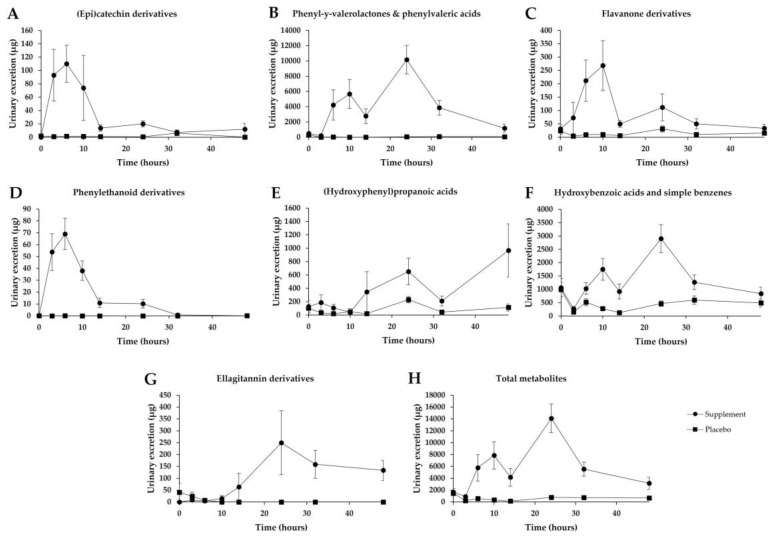
Urinary excretion in placebo (*n* = 10) and supplemented subjects (*n* = 10) over a 48 h period. (**A**) (Epi)catechin derivatives; (**B**) Phenyl-y-valerolactones and phenylvaleric acids; (**C**) Flavanone derivatives; (**D**) Phenylethanoid derivatives; (**E**) (Hydroxyphenyl)propanoic acids; (**F**) Hydroxybenzoic acids and simple benzenes; (**G**) Ellagitannin derivatives; (**H**) Total metabolites. Data are means ± SEM.

**Figure 7 nutrients-14-04913-f007:**
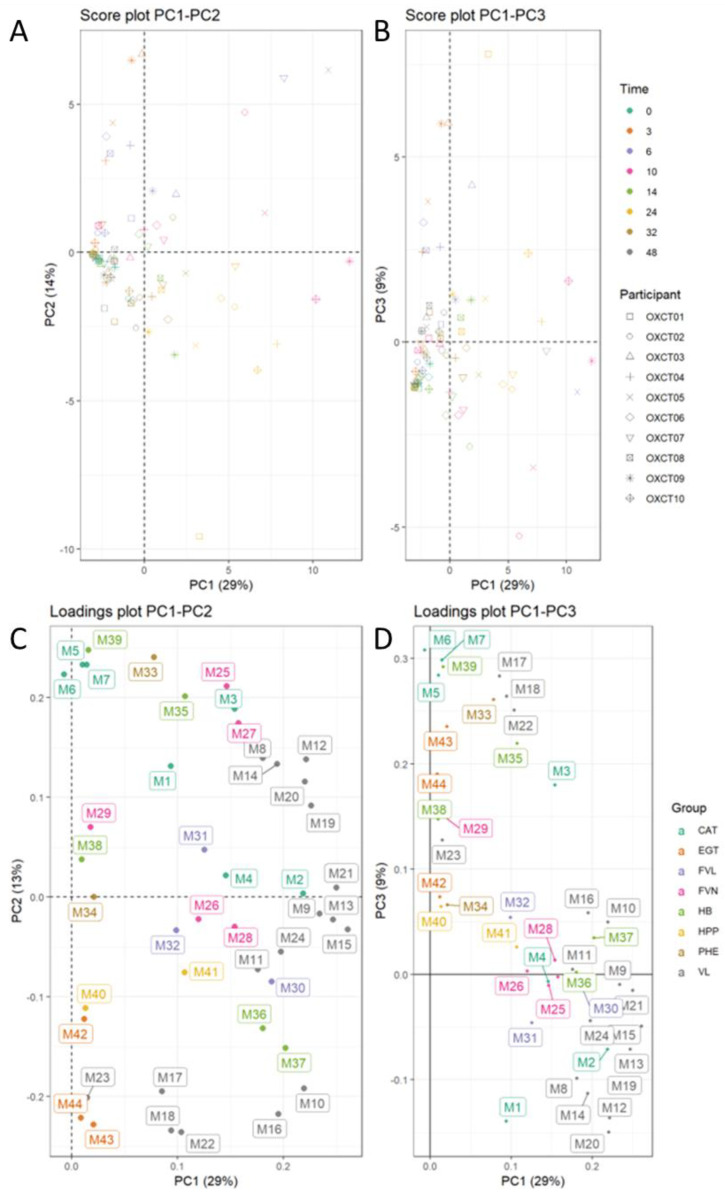
Inter-individual variability for urinary metabolites illustrated through principal component analysis. (PCA). (**A**) Score plots of PC1 versus PC2 for all the volunteers and (**B**) Score plots of PC1 versus PC3; (**C**) Loading plots of PC2 versus PC1 for all urinary metabolites and (**D**) PC1 versus PC3.

**Table 1 nutrients-14-04913-t001:** Comparison of total (poly)phenol content (mg GAE) and antioxidant capacity between 400 g of five servings of fruits and vegetables and a daily dose of 450 mg of the test supplement. Values are reported as mean ± SD (*n* = four different production batches of the supplement).

	Five Fruits and Vegetables—400 g	Supplement—450 mg
Total (poly)phenol content (mg GAE)	228 ± 23	230 ± 23
Total antioxidant capacity (mg GAE)	396 ± 20	383 ± 19

**Table 2 nutrients-14-04913-t002:** Internal HPLC analysis and quantification of the main (poly)phenolic families occurring in the supplement. Values are reported as mean (mg/450 mg) ± SD (*n* = four different production batches of the supplement).

Family	Mean (mg/450 mg)
Flavan-3-ols	50.7 ± 3.2
Flavanones	39.1 ± 2.9
Ellagitannins	26.7 ± 2.0
Secoiridois and Phenylethanoids	22.2 ± 0.9
Anthocyanins	6.1 ± 1.0
**TOTAL (poly)phenols**	**144.8 ± 9.9**

**Table 3 nutrients-14-04913-t003:** Quantification of detected (poly)phenols occurring in the supplement. Values are expressed as mean ± SD (*n* = 3).

Compounds	Mean (µg/450 mg)
**Flavan-3-ols**	
(+)-Catechin	4100.4 ± 169.5
(−)-Epicatechin	4737.9 ± 69.0
(+)-Gallocatechin	399.9 ± 5.9
(−)-Epigallocatechin	1989.7 ± 67.6
(Epi)catechin gallate	28,879.6 ± 639.5
(Epi)gallocatechin-methylgallate	294.6 ± 3.6
(−)-Epigallocatechin gallate	27,109.5 ± 587.8
(Epi)gallocatechin gallate	1516.6 ± 76.1
Procyanidin dimer A-type	4441.0 ± 363.0
Procyanidin dimer B-type	571.7 ± 33.7
Procyanidin dimer B-type	914.9 ± 47.7
Procyanidin dimer B-type	432.5 ± 25.9
Procyanidin dimer B-type	276.0 ± 5.1
Prodelphinidin dimer B-type (1 unit of (epi)GC + 1(epi)C)	70.7 ± 2.2
Prodelphinidin dimer B-type (1 unit of (epi)GC + 1(epi)C)	115.7 ± 3.7
Prodelphinidin dimer B-type gallate (1 unit of (epi)GC + 1(epi)C)	1339.0 ± 96.7
Prodelphinidin tetramer B-type gallate (2 unit of (epi)GC + 2(epi)C)	303.2 ± 14.9
Prodelphinidin tetramer B-type gallate (2 unit of (epi)GC + 2(epi)C)	257.0 ± 48.6
Prodelphinidin dimer B-type gallate (2 units (epi)GC)	83.6 ± 3.6
Procyanidin trimer B-type	n.q.
Procyanidin trimer B-type	n.q.
Procyanidin trimer B-type	n.q.
(Epi)catechin-gallate dimer	n.q.
**Sum of Flavan-3-ols (mg/450 mg)**	**77.83 ± 0.91**
**Ellagitannins**	
Ellagic acid	293.9 ± 8.5
Galloyl-hexoside	1.7 ± 0.3
Galloyl-hexoside	10.9 ± 0.4
Galloyl-hexoside	27.2 ± 1.2
Galloyl-hexoside	21.6 ± 1.5
Galloyl-hexoside	22.3 ± 0.8
Ellagic acid hexoside	116.6 ± 9.1
Ellagic acid dimethyl ether glucurionide	5.3 ± 1.0
Galloyl-HHDP-hexoside	69.9 ± 2.2
Galloyl-HHDP-hexoside	4.5 ± 0.6
Galloyl-HHDP-hexoside	49.5 ± 3.0
Galloyl-HHDP-hexoside	55.4 ± 4.6
Galloyl-HHDP-hexoside	88.2 ± 1.6
Galloyl-HHDP-hexoside	360.7 ± 27.8
Galloyl-HHDP-hexoside	252.0 ± 9.3
Gallagyl-hexoside (Punicalin α/A)	93.1 ± 2.3
Gallagyl-hexoside (Punicalin β/B)	1266.9 ± 156.5
Bis-HHDP-hexoside (Pedunculagin I isomer)	41.0 ± 0.5
Bis-HHDP-hexoside (Pedunculagin I isomer)	249.7 ± 10.2
Bis-HHDP-hexoside (Pedunculagin I isomer)	238.7 ± 11.7
Galloyl-bis-HHDP-hexoside (Casuarinin)	145.0 ± 9.3
Punicalagin isomer	1026.7 ± 59.8
HHDP-gallagyl-hexoside (Punicalagin α)	10,935.5 ± 1237.4
HHDP-gallagyl-hexoside (Punicalagin β)	24,246.1 ± 1017.9
Di(HHDP-galloylglucose)-pentoside	2029.8 ± 182.0
Di(HHDP-galloylglucose)-pentoside	181.8 ± 12.0
Di(HHDP-galloylglucose)-pentoside	662.2 ± 42.6
**Sum of Ellagitannins (mg/450 mg)**	**42.50 ± 2.22**
**Phenylethanoids**	
Tyrosol (tentative identification)	159.0 ± 8.7
Hydroxytyrosol	21,065.2 ± 865.2
Hydroxytyrosol-*O*-hexoside	10,467.2 ± 367.0
Oleoside	517.1 ± 12.0
Verbascoside (Caffeoyl-phenylethanoid glycoside)	488.9 ± 15.0
Verbascoside (Caffeoyl-phenylethanoid glycoside)	81.8 ± 4.1
**Sum of Phenylethanoids (mg/450 mg)**	**32.78 ± 1.25**
**Flavanones**	
Naringenin	56.6 ± 2.2
Eriodyctiol	9.0 ± 0.5
Tetrahydroxyflavanone	6.6 ± 0.2
Naringenin-*O*-glucoside	7.4 ± 0.8
Tetrahydroxyflavanone-*O*-rhamnoside (tentative identification)	22.2 ± 0.3
Hesperetin-*O*-hexoside	681.5 ± 19.9
Naringenin-7-*O*-rutinoside (Narirutin)	427.8 ± 4.2
Naringenin-7-*O*-neohesperidoside (Naringin)	17,923.8 ± 297.3
Naringenin-*O*-neohesperidoside	69.9 ± 1.3
Isosakuranetin-*O*-rutinoside (Didymin)	739.6 ± 6.9
Eriocitrin or Neoeriocitrin	n.q.
Naringenin-*C*-dihexoside	37.9 ± 1.2
Hesperetin-7-*O*-rutinoside (Hesperidin)	7734.6 ± 635.6
Hesperetin-7-*O*-neohesperidoside (Neohesperidin)	90.6 ± 7.3
**Sum of Flavanones (mg/450 mg)**	**27.81 ± 0.95**
**Seco-iridoids**	
Oleuropein aglycone	n.q.
Oleuropein aglycone	n.q.
Oleuropein aglycone	n.q.
Oleuropein aglycone	n.q.
Oleuropein aglycone	n.q.
Oleuropein	15,980.1 ± 1067.3
Oleuropein	3930.0 ± 200.7
**Sum of Seco-iridoids (mg/450 mg)**	**19.91 ± 1.27**
**Flavones**	
Apigenin	n.q.
Trihydroxyflavone	221.7 ± 2.7
Apigenin-*O*-hexoside	100.4 ± 2.8
Luteolin-*O*-hexoside	597.1 ± 18.5
Luteolin-*O*-hexoside	3465.2 ± 142.5
Luteolin-*O*-hexoside	1420.6 ± 62.3
Luteolin-*O*-hexoside	509.2 ± 4.3
Dihydroluteolin-*O*-hexoside	264.7 ± 3.3
Apigenin-*O*-rutinoside	239.2 ± 2.5
Luteolin-*O*-rutinoside	297.7 ± 4.5
Luteolin-*O*-rutinoside	207.9 ± 8.2
Chrysoeriol-*O*-rutinoside or Diosmetin-*O*-rutinoside	35.3 ± 0.4
Luteolin-*O*-dihexoside (tentative identification)	50.5 ± 7.4
Luteolin-*O*-dihexoside	32.0 ± 1.1
**Sum of Flavones (mg/450 mg)**	**7.44 ± 0.10**
**Flavonols**	
Quercetin	3448.8 ± 256.9
Rhamnetin	1.7 ± 0.1
Isorhamnetin	8.0 ± 0.4
Myricetin	7.5 ± 0.4
Quercetin-*O*-pentoside	67.8 ± 2.5
Myricetin-*O*-rhamnoside	115.4 ± 11.3
Quercetin-*O*-hexoside	428.0 ± 10.2
Quercetin-*O*-glucuronide	706.1 ± 16.5
Myricetin-*O*-hexoside	330.5 ± 5.9
Kaempferol-*O*-rutinoside	86.4 ± 4.2
Tetrahydroxy-dimethoxyflavone-*O*- hexoside (Syringetin-*O*-hexoside)	70.6 ± 1.6
Isorhamnetin-*O*- dirhamnoside	225.6 ± 6.5
Quercetin-*O*-rutinoside (Rutin)	187.4 ± 8.6
Isorhamnetin-*O*-rutinoside	106.6 ± 1.6
**Sum of Flavonols (mg/450 mg)**	**5.79 ± 0.25**
**Hydroxybenzoic acids**	
3-Hydroxybenzoic acid	365.5 ± 25.9
4-Hydroxybenzoic acid	191.7 ± 4.5
Hydroxybenzoic acid	349.5 ± 12.1
3,4-Dihydroxybenzoic acid (Protocatechuic acid)	152.9 ± 7.3
Dihydroxybenzoic acid	98.9 ± 7.7
Dihydroxyphenylacetic acid	110.1 ± 34.6
Dihydroxyphenylacetic acid	186.3 ± 26.8
Dihydroxyphenylacetic acid	183.3 ± 9.0
Gallic acid	1804.3 ± 38.2
Hydroxyphenyllactic acid	n.q.
Ethyl-gallate	415.5 ± 11.2
**Sum of Hydroxybenzoic acids (mg/450 mg)**	**3.81 ± 0.01**
**Anthocyanins**	
Cyanidin-3-*O*-hexoside	392.9 ± 5.1
Malvidin-3-*O*-arabinoside	39.8 ± 0.3
Peonidin-3-*O*-hexoside	156.6 ± 7.1
Delphinidin 3-*O*-hexoside	370.6 ± 24.6
Petunidin-3-*O*-hexoside	251.6 ± 5.4
Malvidin-3-*O*-hexoside	285.4 ± 11.4
Cyanidin-3-*O*-rutinoside	87.3 ± 0.9
Delphinidin 3-*O*-rutinoside	106.5 ± 3.4
**Sum of Anthocyanins (mg/450 mg)**	**1.69 ± 0.05**
**Hydroxycinnamic acids**	
Caffeic acid	206.6 ± 2.0
5-Caffeoylquinic acid	382.7 ± 8.6
Ferulic acid-*O*-hexoside	26.4 ± 1.5
**Sum of Hydroxycinnamic acids (mg/450 mg)**	**0.62 ± 0.01**
**Stilbenoids**	
Piceid (Resveratrol-*O*-glucoside)	514.2 ± 13.6
**Sum of Stilbenoids (mg/450 mg)**	**0.51 ± 0.01**
**Gallotannins**	
Digalloylglucose	82.9 ± 1.8
Digalloylglucose	6.5 ± 0.1
Digalloylglucose	27.5 ± 0.9
Digalloylglucose	49.3 ± 4.9
Digalloylglucose	33.1 ± 0.3
Trigalloylglucose	112.7 ± 1.4
**Sum of Gallotannins (mg/450 mg)**	**0.31**
**Dihydrochalcones**	
Phloretin	32.0 ± 0.3
**Sum of Dihydrochalcones (mg/450 mg)**	**0.03 ± 0.01**
**Coumarin**	
Scopoletin-*O*-hexoside	n.q.
**Sum of Coumarin (mg/450 mg)**	**n.q.**
**TOTAL (POLY)PHENOL**	**221.1 ± 9.5 (mg/450 mg)** **594.9** **±** **14.5 (** **µ** **mol/450 mg)**

HHDP: means hexahydroxydiphenoyl; (epi)GC means (epi)gallocatechin; (epi)C means (epi)catechin; n.q means compounds that were identified but not quantified because <LOQ.

**Table 4 nutrients-14-04913-t004:** Pharmacokinetic parameters of phenolic metabolites detected in plasma of the supplemented volunteers (*n* = 10). Values are expressed as mean ± SEM. T_max_: time to reach the maximum plasma concentration; C_max_: maximum plasma concentration; AUC0-24: area under the curve; T_pres_: time of presence of metabolites in circulation (calculated by taking the time interval between the first appearance and the last appearance of the considered metabolite); CV: Coefficient of variation, expressed in percentage.

Id.	Phenolic Metabolites	AUC0-24 (CV) (nmol/L·h)	T_pres_ (h)	T_max_ (h)	C_max_ (nmol/L)
	**(Epi)catechin derivatives**				
5	(Epi)catechin-sulfate_isomer 1	18.41 ± 4.09 (70)	6.65 ± 2.03	8.15 ± 2.69	4.72 ± 0.96
4	(Epi)catechin-glucuronide_isomer 2	43.70 ± 6.66 (48)	10.30 ± 2.29	3.60 ± 0.91	10.88 ± 2.44
3	Methoxy(epi)gallocatechin glucuronide	9.57 ± 2.70 (89)	4.86 ± 1.64	3.14 ± 1.20	4.72 ± 0.62
	**Sum of (Epi)catechin derivatives**	**71.68 ± 9.08 (40)**	**13.35 ± 2.53**	**3.10 ± 0.87**	**16.37 ± 2.13**
	**Flavanone derivatives**				
26	Hesperetin-diglucuronide	5.09 ± 1.32 (82)	6.17 ± 2.04	10.78 ± 2.55	1.49 ± 0.37
27	Naringenin-glucuronide	339.74 ± 75.66 (70)	14.70 ± 2.57	5.30 ± 0.98	95.42 ± 29.73
28	Hesperetin-7-glucuronide	152.05 ± 27.84 (58)	17.83 ± 1.54	7.11 ± 0.90	19.95 ± 2.31
29	Hesperetin-sulfate	173.05 ± 55.23 (101)	15.35 ± 1.92	6.50 ± 0.86	31.62 ± 11.38
	**Sum of flavanone derivatives**	**669.93 ± 98.61 (47)**	**21.65 ± 0.42**	**5.30 ± 1.17**	**124.75 ± 39.01**
	**Other flavonoid derivatives**				
48	Luteolin-sulfate	3.41 ± 0.67 (62)	7.60 ± 2.19	1.55 ± 0.24	0.44 ± 0.14
	**Phenylethanoid derivatives**				
33	2-(Phenyl)ethanol-3′-glucuronide (Hydroxytyrosol-glucuronide)	50.92 ± 5.85 (36)	8.45 ± 2.19	1.45 ± 0.32	24.94 ± 5.63
	**Phenyl-y-valerolactones and phenyl valeric acids**				
20	5-(Methoxyhydroxyphenyl)-γ-valerolactone-sulfate	31.12 ± 8.29 (84)	16.25 ± 1.44	8.25 ± 0.80	7.75 ± 2.52
9	5-(Methoxy-hydroxyphenyl)-γ-valerolactone- glucuronide	7.63 ± 2.52 (105)	9.00 ± 3.06	10.00 ± 3.15	1.54 ± 0.35
16	5-(5′-Hydroxyphenyl)-γ-valerolactone-3′-sulfate	30.46 ± 11.85 (123)	10.57 ± 2.72	8.88 ± 2.41	9.97 ± 4.31
21	5-(3′-Hydroxyphenyl)-γ-valerolactone-4′-sulfate	141.70 ± 50.37 (112)	12.15 ± 1.84	7.00 ± 0.75	40.05 ± 16.85
10	5-(5′-Hydroxyphenyl)-γ-valerolactone-3′-glucuronide	125.55 ± 47.68 (120)	14.44 ± 2.42	12.78 ± 2.85	14.96 ± 5.61
13	5-(Phenyl)-γ-valerolactone-sulfate-glucuronide	10.24 ± 3.34 (103)	10.17 ± 2.42	8.67 ± 2.17	1.55 ± 0.29
22	5-(Methoxyphenyl)-γ-valerolactone-sulfate	18.59 ± 8.47 (144)	7.13 ± 2.33	6.50 ± 1.28	7.34 ± 4.47
23	5-Phenyl-γ-valerolactone-3′-sulfate	10.11 ± 3.89 (122)	9.33 ± 2.45	8.22 ± 2.13	2.15 ± 0.48
17	5-(Phenyl)-γ-valerolactone-3′-glucuronide	22.00 ± 6.41 (92)	7.13 ± 2.27	5.88 ± 0.95	6.34 ± 1.66
15	4-Hydroxy-5-(hydroxyphenyl)valeric acid-sulfate	61.35 ± 23.81 (123)	13.06 ± 2.97	6.19 ± 1.06	13.10 ± 3.94
18	4-Hydroxy-5-(phenyl)valeric acid-sulfate	47.47 ± 18.41 (123)	10.30 ± 2.28	5.60 ± 0.88	11.73 ± 2.95
14	5-(Methoxyphenyl)valeric acid-glucuronide	39.35 ± 14.74 (118)	12.06 ± 2.47	7.22 ± 0.91	8.80 ± 3.60
	**Sum of Phenyl-y-valerolactones and phenyl valeric acids**	**545.58 ± 155.32 (90)**	**21.60 ± 0.72**	**8.50 ± 2.30**	**84.55 ± 36.40**
	**(Hydroxyphenyl)propanoic acids**				
41	3-(4′-hydroxyphenyl)propanoic acid-3′-sulfate (Dihydrocaffeic acid-sulfate)	27.41 ± 11.21 (129)	9.06 ± 2.56	4.56 ± 1.13	18.24 ± 11.27
46	3-(3′-Methoxyphenyl)propanoic acid-4′-sulfate (Dihydroferulic acid-sulfate)	558.28 ± 531.21 (301)	7.14 ± 3.61	6.43 ± 2.97	168.73 ± 156.13
40	3-(3′-Hydroxyphenyl)propanoic acid (3-(3-Hydroxyphenyl)propionic acid)	1554.29 ± 644.33 (131)	7.50 ± 2.38	7.13 ± 0.77	443.09 ± 99.28
	**Sum of (hydroxyphenyl)propanoic acids**	**2139.98 ± 953.73 (141)**	**12.60 ± 2.76**	**5.50 ± 1.01**	**425.53 ± 122.90**
	**Hydroxybenzoic acids and simple benzenes**				
45	4-Hydroxybenzoic acid-3-sulfate (Protocatechuic acid-3-sulfate)	196.06 ± 32.72 (53)	13.45 ± 2.80	2.10 ± 0.46	43.60 ± 7.34
47	4′-Hydroxyhippuric acid	5719.44 ± 1327.87 (73)	21.55 ± 0.74	6.50 ± 1.28	816.10 ± 269.47
36	Dihydroxybenzene-sulfate (Pyrogallol-sulfate)	1677.03 ± 322.08 (61)	23.30 ± 0.27	7.10 ± 0.86	185.03 ± 34.67
37	Methoxyhydroxybenzene-sulfate (Pyrogallol-methoxy-sulfate)	885.81 ± 266.69 (95)	21.35 ± 1.41	7.00 ± 0.93	108.13 ± 24.62
	**Sum of hydroxybenzoic acids and simple benzenes**	**8478.34 ± 1448.99 (54)**	**23.60 ± 0.22**	**7.70 ± 0.86**	**958.18 ± 269.69**
	**SUM of all metabolites**	**11,959.85 ± 2200.65 (58)**	**23.80 ± 0.24**	**6.90 ± 0.96**	**1271.11 ± 281.33**

**Table 5 nutrients-14-04913-t005:** Phenolic metabolites detected in urine of the supplemented volunteers (*n* = 10). Values are expressed as µg of compound, and data are reported as mean ± SEM. nd means not detected.

Id.	Phenolic Metabolites	0 (h)	0–3 (h)	3–6 (h)	6–10 (h)	10–14 (h)	14–24 (h)	24–32 (h)	32–48 (h)	Total (0–48 h)	CV (%)
	**(Epi)catechin derivatives**										
5	(Epi)catechin-sulfate_isomer 1	nd	14.26 ± 7.60	10.63 ± 3.02	0.49 ± 0.49	nd	nd	nd	2.28 ± 2.28	27.65 ± 8.42	96
6	(Epi)catechin-sulfate_isomer 2	nd	36.14 ± 15.62	29.95 ± 11.99	0.00 ± 0.00	1.66 ± 1.20	4.12 ± 2.29	2.06 ± 2.06	9.52 ± 6.72	83.45 ± 19.91	75
2	(Epi)catechin-glucuronide_isomer 1	nd	0.17 ± 0.17	2.60 ± 1.15	9.09 ± 3.87	3.10 ± 1.10	5.67 ± 1.80	1.72 ± 0.92	nd	22.35 ± 5.67	80
4	(Epi)catechin-glucuronide_isomer 2	nd	0.59 ± 0.59	14.00 ± 7.73	53.64 ± 44.88	3.19 ± 1.65	3.65 ± 2.21	1.22 ± 1.22	nd	76.29 ± 45.22	187
7	Methoxy(epi)catechin-sulfate	nd	39.90 ± 19.16	48.81 ± 12.64	7.02 ± 4.78	4.33 ± 2.25	5.00 ± 3.44	2.12 ± 2.12	nd	107.19 ± 26.64	79
1	Methoxy-(epi)catechin-glucuronide	nd	nd	0.59 ± 0.52	1.29 ± 0.94	0.74 ± 0.36	0.24 ± 0.22	nd	nd	2.86 ± 1.48	164
3	Methoxy-(epi)gallocatechin-glucuronide	nd	1.91 ± 0.99	3.38 ± 0.81	2.36 ± 1.11	0.78 ± 0.33	1.33 ± 0.61	0.18 ± 0.18	nd	9.94 ± 2.24	71
	**Sum of (Epi)catechin derivatives**	**nd**	**92.97 ± 38.65**	**109.96 ± 27.97**	**73.89 ± 48.67**	**13.80 ± 5.01**	**20.00 ± 4.58**	**7.30 ± 3.54**	**11.80 ± 8.79**	**329.72 ± 86.05**	**83**
	**Flavanone derivatives**										
29	Hesperetin-sulfate	nd	2.03 ± 2.03	1.19 ± 0.81	0.23 ± 0.23	nd	1.22 ± 0.90	0.33 ± 0.33	2.21 ± 2.21	7.21 ± 5.41	237
28	Hesperetin-7-glucuronide	14.48 ± 5.94	4.02 ± 2.90	20.07 ± 4.27	28.56 ± 5.46	9.57 ± 2.19	30.72 ± 8.04	19.07 ± 5.57	22.33 ± 5.83	148.81 ± 14.88	32
26	Hesperetin-diglucuronide	nd	0.05 ± 0.05	0.16 ± 0.08	0.12 ± 0.07	0.10 ± 0.05	0.45 ± 0.26	0.16 ± 0.07	0.04 ± 0.04	1.09 ± 0.36	104
27	Naringenin-glucuronide	16.31 ± 16.31	65.92 ± 53.49	187.75 ± 72.43	237.98 ± 90.88	39.21 ± 11.90	78.88 ± 45.12	30.40 ± 16.54	9.13 ± 9.13	665.58 ± 148.66	71
25	Naringenin-diglucuronide	0.17 ± 0.17	0.56 ± 0.38	2.47 ± 1.02	1.66 ± 1.01	0.46 ± 0.21	0.54 ± 0.28	0.09 ± 0.09	nd	5.95 ± 1.56	83
	**Sum of flavanone derivatives**	**30.96 ± 15.78**	**72.59 ± 58.76**	**211.64 ± 77.44**	**268.55 ± 93.64**	**49.33 ± 12.85**	**111.81 ± 50.11**	**50.04 ± 19.22**	**33.70 ± 15.01**	**828.64 ± 159.16**	**61**
	**Other flavonoid derivatives**										
30	Luteolin-glucuronide	nd	1.84 ± 1.25	2.67 ± 0.96	3.39 ± 2.32	1.80 ± 0.84	5.97 ± 2.77	3.07 ± 2.06	6.12 ± 2.55	24.88 ± 7.86	100
31	Myricetin-glucuronide	nd	0.00 ± 0.00	0.72 ± 0.49	2.78 ± 1.05	1.81 ± 0.68	3.68 ± 1.04	1.30 ± 0.56	nd	10.29 ± 2.19	67
32	Quercetin-diglucuronide	nd	0.04 ± 0.04	0.39 ± 0.19	0.50 ± 0.25	0.21 ± 0.19	0.42 ± 0.25	0.30 ± 0.23	nd	1.86 ± 1.01	171
	**Sum of other flavonoid derivatives**	**nd**	**1.88 ± 1.29**	**3.78 ± 1.42**	**6.68 ± 2.59**	**3.83 ± 1.14**	**10.07 ± 2.88**	**4.67 ± 2.16**	**6.12 ± 2.55**	**37.03 ± 7.60**	**65**
	**Phenylethanoid derivatives**										
34	Oleuropein-sulfate	0.02 ± 0.02	0.01 ± 0.01	0.03 ± 0.01	nd	nd	0.01 ± 0.01	0.03 ± 0.01	0.02 ± 0.02	0.10 ± 0.03	106
33	2-(Phenyl)ethanol-3′-glucuronide (Hydroxytyrosol-glucuronide)	nd	53.75 ± 15.51	69.04 ± 13.20	38.02 ± 8.17	10.90 ± 3.90	10.23 ± 3.71	0.75 ± 0.75	nd	182.69 ± 19.99	35
	**Sum of phenylethanoid derivatives**	**0.02 ± 0.02**	**53.76 ± 15.51**	**69.06 ± 13.20**	**38.02 ± 8.17**	**10.90 ± 3.90**	**10.24 ± 3.72**	**0.77 ± 0.74**	**0.02 ± 0.02**	**182.79 ± 19.98**	**35**
	**Phenyl-y-valerolactones and phenylvaleric acids**										
8	5-(Dihydroxyphenyl)-γ-valerolactone-glucuronide	nd	nd	23.24 ± 17.36	26.86 ± 11.59	6.17 ± 3.84	5.33 ± 2.23	nd	nd	61.61 ± 29.59	152
20	5-(Methoxy-hydroxyphenyl)-γ-valerolactone--sulfate	nd	1.15 ± 0.85	123.73 ± 64.97	287.71 ± 81.19	98.28 ± 40.32	116.82 ± 44.46	21.40 ± 9.99	2.02 ± 2.02	651.12 ± 170.13	83
9	5-(Methoxyhydroxyphenyl)-γ-valerolactone- glucuronide	nd	nd	9.83 ± 6.55	18.07 ± 6.62	9.28 ± 2.54	16.44 ± 4.84	4.47 ± 2.31	nd	58.09 ± 11.33	62
16	5-(5′-Hydroxyphenyl)-γ-valerolactone-3′-sulfate	18.25 ± 10.84	4.64 ± 3.47	439.69 ± 219.42	1256.55 ± 516.93	944.18 ± 474.02	3986.28 ± 711.64	1721.60 ± 393.97	586.15 ± 240.93	8957.33 ± 1361.49	48
21	5-(3′-Hydroxyphenyl)-γ-valerolactone-4′-sulfate	344.60 ± 186.50	166.29 ± 126.77	2548.76 ± 1207.06	2277.16 ± 795.52	919.49 ± 314.48	3618.13 ± 985.44	1346.37 ± 481.02	389.78 ± 215.48	11,610.60 ± 1923.81	52
10	5-(5′-Hydroxyphenyl)-γ-valerolactone-3′-glucuronide	5.34 ± 5.34	0.00 ± 0.00	101.70 ± 51.36	316.86 ± 132.04	199.53 ± 88.98	624.93 ± 108.36	281.58 ± 60.75	65.29 ± 25.62	1595.23 ± 260.71	52
13	5-(Phenyl)-γ-valerolactone-sulfate-glucuronide	0.34 ± 0.34	0.13 ± 0.13	9.26 ± 4.10	23.29 ± 7.81	10.14 ± 3.00	20.29 ± 4.37	7.77 ± 2.43	1.43 ± 0.97	72.65 ± 15.10	66
22	5-(Methoxy-phenyl)-γ-valerolactone-sulfate	5.66 ± 3.78	2.16 ± 1.13	7.50 ± 2.92	18.92 ± 9.27	12.82 ± 4.86	22.47 ± 5.87	9.17 ± 3.61	4.24 ± 2.18	82.94 ± 15.05	57
23	5-(Phenyl)-γ-valerolactone-3′-sulfate	41.07 ± 30.35	4.29 ± 2.19	52.98 ± 22.54	175.24 ± 98.80	76.01 ± 39.48	345.14 ± 198.47	100.92 ± 46.88	34.61 ± 16.03	830.26 ± 275.43	105
17	5-(Phenyl)-γ-valerolactone-3′-glucuronide	15.01 ± 15.01	15.42 ± 15.42	196.48 ± 138.99	169.91 ± 105.95	65.96 ± 36.27	468.44 ± 330.34	81.71 ± 28.26	16.56 ± 16.56	1029.49 ± 394.77	121
15	4-Hydroxy-5-(Hydroxyphenyl)valeric acid-sulfate	31.48 ± 18.11	9.56 ± 5.01	268.65 ± 125.49	432.54 ± 133.32	216.01 ± 58.49	571.58 ± 137.24	216.26 ± 60.42	30.47 ± 14.26	1776.56 ± 310.38	55
19	4-Hydroxy-5-(methoxy-phenyl)valeric acid--sulfate	nd	0.30 ± 0.21	3.34 ± 1.93	6.42 ± 1.94	1.84 ± 0.84	4.02 ± 1.53	0.29 ± 0.15	nd	16.20 ± 4.31	84
11	4-Hydroxy-5-(methoxyphenyl)valeric acid-glucuronide (4-Hydroxy-5-phenylvaleric acid-methoxy-glucuronide)	nd	nd	4.79 ± 2.48	6.48 ± 4.24	8.52 ±3.93	18.97 ± 6.71	1.78 ±1.78	nd	40.54 ± 10.11	79
18	4-Hydroxy-5-(phenyl)valeric acid-sulfate	0.81 ± 0.61	0.23 ± 0.11	2.28 ± 1.17	4.40 ± 2.19	2.01 ± 0.97	10.89 ± 6.29	3.86 ± 1.36	1.41 ± 0.57	25.89 ± 8.19	100
24	5-(Hydroxyphenyl)valeric acid-sulfate	5.68 ± 5.68	0.58 ± 0.42	5.06 ± 5.06	7.41 ± 4.74	4.35 ± 2.69	34.54 ± 13.80	31.20 ± 5.27	29.85 ± 15.45	118.67 ± 27.65	74
12	5-(Methoxyphenylvaleric) acid-sulfate	nd	1.90 ± 1.90	374.69 ± 221.67	543.24 ± 149.94	169.00 ± 78.02	277.94 ± 104.02	42.67 ± 21.21	5.27 ± 3.42	1414.71 ± 421.96	94
14	5-(Methoxy-phenylvaleric) acid--glucuronide	8.97 ± 6.01	nd	59.80 ± 41.17	90.37 ± 34.08	22.28 ± 11.86	14.60 ± 7.48	nd	3.07 ± 3.07	199.09 ± 78.07	124
	**Sum of phenyl-y-valerolactones and phenylvaleric acids**	**477.22 ± 266.28**	**206.65 ± 148.90**	**4231.78 ± 1978.16**	**5661.43 ± 1887.41**	**2765.88 ± 963.32**	**10,156.83 ± 1864.13**	**3871.06 ± 972.99**	**1170.14 ± 510.79**	**28,540.99 ± 3805.96**	**42**
	**(Hydroxyphenyl)propanoic acids**										
41	3-(4′-hydroxyphenyl)propanoic acid-3′-sulfate (Dihydrocaffeic acid-sulfate)	27.49 ± 27.49	101.08 ± 52.47	103.59 ± 54.34	41.08 ± 16.51	56.67 ± 24.78	389.70 ± 95.00	208.69 ± 73.47	289.45 ± 118.16	1217.76 ± 256.95	67
40	3-(3′-Hydroxyphenyl)propanoic acid (3-(3-Hydroxyphenyl)propionic acid)	96.20 ± 96.20	85.67 ± 85.67	nd	nd	287.94 ± 287.94	262.06 ± 138.26	nd	676.70 ± 383.40	1408.57 ± 544.97	122
	**Sum of (Hydroxyphenyl)propanoic acids**	**123.69 ± 97.07**	**186.75 ± 112.79**	**103.59 ± 54.34**	**41.08 ± 16.51**	**344.61 ± 305.36**	**651.76 ± 199.89**	**208.69 ± 73.47**	**966.15 ± 399.30**	**2626.33 ± 618.94**	**75**
	**Hydroxybenzoic acids and simple benzenes**										
39	3,5-Dimethoxy-4-hydroxybenzoic acid (Syringic acid)	0.71 ± 0.46	8.71 ± 8.71	3.89 ± 3.81	3.57 ± 2.82	1.42 ± 1.38	3.63 ± 2.95	3.33 ± 2.94	8.73 ± 8.35	33.92 ± 30.72	286
38	Methoxy-hydroxybenzoic acid-sulfate (Gallic acid-methoxy-sulfate)	1.18 ± 1.18	18.56 ± 8.39	26.07 ± 5.72	2.89 ± 1.25	1.18 ± 1.18	2.43 ± 1.72	2.23 ± 1.49	2.34 ± 2.34	56.87 ± 13.94	78
35	4-Hydroxybenzoic acid-3-glucuronide (Protocatechuic acid-3-glucuronide)	1.13 ± 1.13	14.20 ± 4.88	24.68 ± 5.82	12.53 ± 4.10	3.10 ± 1.12	4.51 ± 1.34	nd	nd	60.16 ± 9.86	52
36	Dihydroxybenzene-sulfate (Pyrogallol-sulfate)	747.18 ± 254.96	125.75 ± 42.49	549.56 ± 129.84	948.39 ± 201.27	504.78 ± 150.35	1676.23 ± 346.95	721.74 ± 169.19	541.99 ± 169.88	5815.62 ± 765.07	42
37	Methoxyhydroxybenzene-sulfate (Pyrogallol-methoxy-sulfate)	311.66 ± 96.06	104.78 ± 28.89	425.64 ± 95.78	783.20 ± 204.13	409.48 ± 127.88	1211.89 ± 191.45	538.08 ± 118.63	284.32 ± 101.86	4069.06 ± 502.55	39
	**Sum of hydroxybenzoic acids and simple benzenes**	**1061.79 ± 329.42**	**272.00 ± 85.28**	**1029.84 ± 219.14**	**1750.58 ± 408.15**	**919.95 ± 279.48**	**2898.70 ± 524.69**	**1265.38 ± 274.37**	**837.38 ± 246.45**	**10,035.62 ± 1234.49**	**39**
	**Ellagitannin derivatives**										
42	8-Hydroxy-urolithin-3-glucuronide (Urolithin A-glucuronide)	nd	8.35 ± 7.92	4.88 ± 4.63	2.42 ± 2.30	18.31 ± 14.57	68.99 ± 33.01	69.22 ± 21.30	96.89 ± 37.45	269.07 ± 84.25	99
43	9-Hydroxy-urolithin-3-glucuronide (Isourolithin A-glucuronide)	nd	nd	nd	6.91 ± 6.55	5.49 ± 5.21	36.41 ± 34.54	14.38 ± 9.83	5.33 ± 5.05	68.53 ± 55.19	255
44	Urolithin-3-glucuronide (Urolithin B-glucuronide)	nd	nd	nd	5.31 ± 5.04	39.56 ± 37.53	144.45 ± 90.90	74.85 ± 48.79	30.99 ± 29.40	295.17 ± 185.26	198
	**Sum of ellagitannin derivatives**	**nd**	**8.35 ± 7.92**	**4.88 ± 4.63**	**14.64 ± 11.53**	**63.37 ± 57.13**	**249.86 ± 135.01**	**158.45 ± 58.48**	**133.21 ± 42.40**	**632.76 ± 251.20**	**126**
	**SUM of all metabolites**	**1693.68 ± 579.94**	**894.12 ± 325.73**	**5764.04 ± 2219.80**	**7853.42 ± 2304.78**	**4165.34 ± 1501.54**	**14,084.29 ± 2394.88**	**5550.52 ± 1194.80**	**3145.19 ± 1052.77**	**43,150.60 ± 5117.69**	**38**

## Data Availability

The data presented in this study are available on request from the corresponding author, due to privacy restriction.

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
