# Peer review of "Exposure to (Poly)phenol Metabolites after a Fruit and Vegetable Supplement Intake: A Double-Blind, Cross-Over, Randomized Trial"

_nutrients, 2022, doi:10.3390/nu14224913_

Round 1

Reviewer 1 Report

Romain and collaborators aimed to validate the proof-of-concept of a (poly)phenolic supplementation based on phenolic compound intake comparable to a “5-a-day” fruit and vegetable recommendation. In general, the experimental approach appears to be suitable to address the research question with appropriate controls and adequate analytical methodology, although there do appear to be some gaps. There are some specific comments related to some specific aspects of the experimental approach which require clarification. In addition, there are some data presentation aspects that also should be addressed.

1.       Line 290, the abbreviation for mass spectrometry has already been introduced on line 142

2.       Lines 256 and 257: leave a space between the numerical value and the unit (1 h, 2 h…)

3.       Line 361: Type error kg/m2, put in superscript 2

4.       Line 402: Throughout the text the symbol ± must be separated from the numerical value, this error is constantly repeated (lines 404, 415, 417, 424………………)

5.       Please improve the quality of Figures 1 and 2

6.       Table 1: indicate the number of replicates

7.       Table 3: it is recommended that the compounds (grouped by families) be listed in the same order as they appear in the text. This will help to better understand the table. It is also suggested that at the end of each family, the total of these compounds should appear (mg/450 mg), which should coincide with the percentage of the compounds

8.       Line 416: be careful when using phrases such as “Only one phenylethanoid derivative, namely 2-(phenyl)ethanol-3′-glucuronide (Id. 416 33), was quantified in the plasma” because perhaps more than one compound could have been detected using HRMS. The reviewer suggests that it is more appropriate to express it as follows Only one phenylethanoid derivative, namely 2-(phenyl)ethanol-3′-glucuronide (Id. 416 33), was quantified in the plasma, with the analytical method employed”

9.       Table 5: In the header, the 3-6 h range is duplicated, please change the second value to 6-10 (h)

10.   Table 5: Why for some compounds there are some values that appear at 0.00 ± 0.00 at intermediate times and then the concentration rises. Examples: Hesperetin-sulfate, Oleuropein-sulfate, 5-(Dihydroxyphenyl)-γ-valerolactone-glucuronide…..

11.   Figure 6: The reviewer suggests presenting excretion graphs by compound family that will facilitate the reader to understand the trend of double excretion peaks, etc.

Reviewer 2 Report

1. The title is too long and needs specificity.

2. Keywords should not repeat words from the title.

3. The abstract needs to state the purpose.

4. There is a lack of comparison of the supplement with commonly consumed fruits and vegetables in terms of phenolic compounds.

5. There is a lack of indication of the limitation of the study, including the economic aspect and the synergistic effect more favorable to health when consuming the full food matrix rather than the supplement.

6. The quality of Figure 7 needs to be improved.

7. "Supplementation of a complex (poly)phenolic ingredient could overcome the issue of bioavailability." - In what way?

8. The manuscript must not be an advertisement for the supplement. The Authors should point out the benefits of fruit and vegetable consumption associated with a full matrix of foods rich not only in polyphenols but also in fiber, vitamins, minerals. 

Round 2

Reviewer 1 Report

I thank the authors for considering my suggestions. The current format of the article seems to me to be suitable for publication.